# The Role of Audit Committee Characteristics and I.C. Performance on I.C. Disclosure: Evidence from the Indonesian Banking Sector

**Wisnu Mawardi \*, Harjum Muharam**  **and Mulyo Haryanto**

Faculty of Economics and Business, Universitas Diponegoro, Semarang 50275, Indonesia
\* Correspondence: wisnumawardi@lecturer.undip.ac.id

**Abstract:** This study aims to analyze the influence of audit committee characteristics and intellectual capital performance on intellectual capital disclosure. Characteristics of the audit used in this research are the size of the audit committee, the Number of audit committee meetings, and the financial expertise of the audit committee. The population in this study is a financial services company listed on the Indonesia Stock Exchange in 2019–2021 and collected a sample of 91 companies using the purposive sampling technique. The analysis method used in this research is multiple linear regression using the software SPSS 20. The test result of this study shows that an audit committee or several audit committees positively affect intellectual capital disclosure. However, at the same time, the financial expertise of the audit committee and intellectual capital performance does not affect intellectual capital disclosure.

**Keywords:** audit committee; intellectual capital performance; intellectual capital disclosure

**JEL Classification:** G32; M14; M42

## 1. Introduction

Companies can disclose intellectual capital through the company's annual financial report, a tool for conveying information to stakeholders. The annual financial report contains two pieces of information that must be disclosed, namely voluntary and mandatory disclosure (Dwipayani and Putri 2016). Mandatory disclosure is information that must be disclosed by a company regulated by the applicable regulatory body. By contrast, regarding the voluntary disclosure information, the company is not obligated to adhere to the required disclosures or applicable regulations. Intellectual capital disclosure is an annual financial report owned by a company that discloses examples of voluntary disclosures. Disclosure of intellectual capital comprises information, intellectual assets, and knowledge. Intellectual capital disclosure can detect opportunities and control threats that can affect the company, impacting the business' resilience and strength in the competition.

Furthermore, disclosure of intellectual capital that is carried out voluntarily by a company can minimize the problems of information asymmetry and potentially harmful influences on the company's reputation and stakeholder trust (Widiatmoko et al. 2020). The regulation stipulates that public companies must report their intellectual capital in their financial statements. Accordingly, this voluntary company's annual report includes this closure of intellectual capital (Hariyanto 2022).

An example was reported on medcom.id (2017) (accessed on 21 June 2022), an exciting piece about the disclosure of intellectual capital regarding P.T. Bank BRI (Persero) Tbk, which was required to resolve severance pay issues by retirees who felt they had not received their proper rights. This problem has been ongoing since 2012 and was resolved in 2013. However, in 2017, when there was a new President Director at BRI, retirees again tried to express their problems. However, in 2013, the problem had been resolved based on

the applicable law. According to Bank BRI, they had completed their obligations to BRI retirees, but the bank needed to disclose the employee cost incurred. This caused a dispute between several parties due to the lack of transparency by Bank BRI. This problem indicates that Bank BRI lacks voluntary disclosure of internal information outside of its financial statements, such as additional information regarding the costs incurred for employees. The goal is that employees have proof that the company has appropriately issued the rights received from the company.

In addition to the case of Bank BRI, there is also a case of theft of customer funds of P.T. Bank Maybank Indonesia Tbk caused by Maybank employees themselves. Reporting to the CNN Indonesia news site (2020), it was stated that one of the customers of Maybank lost IDR 20 billion from his savings account. After being reported to the police, an investigation was carried out. The perpetrator of this burglary was the bank's internal party, the head of the branch. With this case in mind, OJK planned to evaluate Maybank's internal control system related to cases carried out by Maybank employees. Considering this case, customers should know about the internal control system of a bank before they choose to use financial services to hold savings. Therefore, disclosure of intellectual capital that is carried out voluntarily by a company can help customers to determine reliable financial services for storing their funds.

Furthermore, a committee can encourage the level of supervision needed to optimize intellectual capital disclosure. In this case, the audit committee can determine whether the management working on the report has submitted a proper explanation and is in line with the regulations that have been determined. The Financial Services Authority's regulation (number 55/POJK.04/2015) concerning "Formation and Guidelines for the Implementation of the Work of the Audit Committee" states that companies whose names are on the IDX are required to establish an audit committee in addition to determining the characteristics of the audit committee. The performance of intellectual capital also plays a vital role in disclosing intellectual capital. Intellectual capital performance can be measured through innovation and creativity possessed by a company's workers. Management must know its company's performance to create and maintain added value. Therefore, disclosure of intellectual capital provides added value to a company. Better intellectual capital performance is an advantage for a company. The higher the company's performance, the more the company will benefit from the disclosure of intellectual capital that has been made (Gamayuni 2015).

In a case study conducted by (Hariyanto 2022), it was found that the size of the audit committee affected intellectual capital disclosure. Meanwhile, the case study of (Hesniati 2021) supports the results of this study. His research found that the size of the audit committee is an essential element in explaining intellectual capital disclosure. As the Number of committee members increases, the quality control supervision of the company's accounting and financial processes also increases (Ferreira et al. 2012). The results of the (Whiting and Woodcock 2011) research also indicated that the results obtained can increase the company's disclosures. However, there are inconsistent results; in another study (Tulung et al. 2018), there is a committee size that does not significantly impact the disclosure of intellectual capital. This is because the audit committee, which is a function, does not have a structure and composition sourced from the board of commissioners. Therefore, the size of the audit committee tends to be detrimental to the process of diffusion and responsibility. In Indonesia, the Number of audit committee members is regulated by OJK regulations. However, regulations regarding intellectual capital disclosure in Indonesia have yet to be regulated in detail, so only a few companies have disclosed them (Nuzula et al. 2021).

In Hariyanto (2022), the results show an effect on the number of audit committee meetings with intellectual capital disclosure. These results are supported by the research of (Li et al. 2008) with the same results as the previous research. The Number of audit committee meetings encouraged cooperation between members to work effectively when supervising financial statements. Based on meetings held regularly and periodically by members of the audit committee, the audit committee could consider problems in business competition

and the shift from a labour-based to a knowledge-based operation. However, (Whiting and Woodcock 2011) found different results, which found no influence between the Number of meetings on intellectual capital disclosure. This is because the audit committee meetings that were held regularly were used to implement pre-existing regulations; as a result, the number of audit committee meetings cannot be the basis upon which the audit committee had carried out adequate supervision.

The results of research from Hesniati (2021) show a relationship between the financial expertise of audit committees and the disclosure of intellectual capital. The results of this study are supported by further research conducted by (Mawardi et al. 2020; Rimawi et al. 2021) with the same results as Masita and Muslih (2017). The audit committee's financial expertise is a measure of the committee. Therefore, the disclosure expertise can be influenced by the finances of the audit committee in certain companies (Irwandi and Pamungkas 2020). However, the research of Ahmed Ahmed Haji (2015) found the opposite result. Namely, there is no influence between the audit committee's financial expertise and intellectual capital disclosure. To improve the disclosure, the audit committee's financial and accounting expertise needs to be improved. Thus, the audit committee still needs expertise in other fields. Such as management expertise, technology and information, and so on. The same was also found in (Orens et al. 2009). Therefore, financial expertise is less relevant to disclosing intellectual capital but more relevant to discussing finance.

The results (Gamayuni 2015) show no effect of intellectual capital performance on intellectual capital disclosure. This is supported by (Abdulrahman Anam et al. 2011). Namely, the researchers did not find any effect of intellectual capital performance on intellectual capital disclosure. This is because detailed regulatory arrangements regarding intellectual capital disclosure still needed to be created, thus causing management not to increase its disclosure of intellectual capital. However, (Hesniati 2021)'s research found that intellectual capital performance affects intellectual capital disclosure. The better the performance of intellectual capital owned by a company, the higher (good) the likelihood that a company will disclose intellectual capital. In this case, annual financial reports play a role in reporting disclosure (Brüggen et al. 2009).

This study aimed to examine the effect of the size of the audit committee, the Number of audit committee meetings, and the financial expertise of the audit committee on intellectual capital disclosure. In addition, in this study, intellectual capital performance was tested on the disclosure of intellectual capital for companies engaged in services, especially in the part of the financial sector listed on the IDX.

## 2. Literature Review

### 2.1. Intellectual Capital

There is a need to go beyond I.C. reporting. Innovations in ICD, such as integrating reporting, disclosure in ecosystems, and stakeholder engagement, open up new possibilities for future research (Dumay 2012) on updating and reapplying existing approaches to today's dynamic, knowledge-driven, intangible-based organizations (Cuozzo et al. 2017). Comparability across companies, moving beyond a Euro-centric view of I.C. or helping investors find the suitable needles in the haystack of their information overload are critical.

Tan et al. (2007) classify I.C. into three basic formats, namely: (1) human capital; (2) structural capital; and (3) customer capital. (Leliaert et al. 2003) developed the 4-Leaf model, which classifies I.C. into human, customer, structural capital and strategic alliance capital (Tan et al. 2007). Intellectual capital referred to in this study is I.C. performance, which is measured based on the value added created by physical capital (VACA), human capital (VAHU), and structural capital (STVA). The combination of the three added values is symbolized by the name VAIC™, which was developed by (Pulic 1998).

Value added is the most objective indicator for assessing business success and shows a company's ability to create value (value creation). V.A. is calculated as the difference between output and input. Output (OUT) represents revenue and includes all products and services sold on the market, while input (IN) includes all expenses in obtaining revenue.

The critical aspect of this method is that labour expenses are not included in the IN. Due to its active role in the value creation process, intellectual potential (represented by labour expenses) is not counted as a cost and is not included in the IN component. Therefore, an essential aspect of Pulic's method is treating labour as a value-creating entity. V.A. is influenced by the efficiency of human capital (H.C.) and structural capital (S.C.). Another relationship that V.A. has is with capital employed (C.E.), which in this case is labelled VACA. VACA is an indicator for V.A. created by one unit of physical capital.

Intellectual capital is an asset included in intangible assets or assets that do not have a form. According to PSAK 19, intangible assets are non-monetary assets with no physical form that can be identified and also play a role in obtaining or delivering goods or services, which are leased to other parties based on administrative purposes. Intellectual capital is included in intangible assets, which can contribute to improving competitive position, the way that the value of a company is added by interested parties (Gallardo-Vázquez et al. 2019). In intellectual capital, three components cannot be separated: human capital, organizational capital, and relational capital (Bratianu and Orzea 1997). The following is an explanation of these three components of intellectual capital.

(1) Human capital (H.C.) is the lifeblood of intellectual capital and a source of innovation and improvement. However, in making measurements, H.C. is an element challenging to quantify in these stages. Thus, H.C. can be interpreted as employees' knowledge, abilities, knowledge, relationships, and attitudes.

(2) Internal structure (organizational capital) is the ability of a company to support employee performance to be more effective and comprehensive by carrying out several activities routinely in filling out the structure. For example, organizational culture, company operational systems, management philosophy, manufacturing processes, and all forms of intellectual property; the company owns them. Every individual can possess high intellectuality. However, if the organization or company does not have sound systems and procedures, this intellectual capital and potential cannot be achieved and utilized optimally.

(3) External structure (relational capital) is a good relationship between the company and its partners, sourced from suppliers, customers, the government, and the community. External structures arise from outside the company's environment, such as quality suppliers, loyal customers to company services, good relations between the company and the government, and harmonious relationships with the surrounding community.

### 2.2. Intellectual Capital Disclosure

Disclosure of intellectual capital is a resource that can be provided by a concept in the form of new knowledge or knowledge whose contents are intangible values of a company, as well as describing intangible assets to increase the value of the company (Gamayuni 2015). Disclosure of information regarding intellectual capital in a company is intended to meet the overall information needs of report users who do not participate in compiling company reports. With the annual report, it is hoped that they can know the overall condition of the company. Disclosure of non-financial and financial information must be carried out by companies in Indonesia in their annual reports. Bapepam Kep supports this—134/B.L./2006 concerns the obligation to submit reports.

In their annual reports, there are five reasons companies disclose intellectual capital (Ferreira et al. 2012). First, the disclosure of intellectual capital can assist companies in planning business strategies by identifying and developing intellectual capital to gain a competitive advantage. Second, disclosure of intellectual capital can help develop essential indicators related to a company's achievements that help achieve strategies derived from the company's evaluation results. Third, disclosure of intellectual capital can help evaluate mergers and takeovers of companies. Fourth, the information in the disclosure can be related to the company's compensation and intensive planning. Fifth, this disclosure can be used as a communication tool to stakeholders about the intellectual property owned by the company.

In several previous studies, the ICD Framework was developed and used for intellectual capital disclosure, including research by (Abeysekera 2011; Ferreira et al. 2012; Gamayuni 2015; Hesniati 2021; Widiatmoko et al. 2020). In this study, researchers took 22 items used as research criteria. This was to develop a scheme that understands intellectual capital (Sveiby 1997).

### 2.3. The Effect of Audit Committee Characteristics on Intellectual Capital Disclosure

Agency theory states that the different goals of shareholders and the goals of managers can cause conflict. This is because managers usually pursue personal goals (Hariyanto 2022). Disclosure of intellectual capital is needed to avoid this conflict. The audit committee of a company can supervise the company's management. The audit committee can also help the company's management to minimize information asymmetry between managers and shareholders through abundant information disclosure, an element of which is the disclosure of intellectual capital information.

In the theory of stewardship, it is stated that management is afforded trust to work well for the benefit of the public and stakeholders (Li et al. 2008). In this study, the audit committee helps the management (steward) to supervise and provide the best results for the company. A company's intellectual capital runs effectively with an audit committee that works well. This encourages management to disclose more intellectual capital owned by the company. Disclosure of intellectual capital is related to the characteristics of the audit committee. The descriptions below explain the relationship between disclosures and these characteristics. The size of the audit committee, the financial expertise of the audit committee, and the Number of audit meetings are all characteristics of the committee.

Regarding the effect of the size of the audit committee on the disclosure of intellectual capital, POJK No.55/POJK.04/2015 reports that the board of commissioners forms the audit committee. The audit committee assists the board of commissioners when carrying out their duties and functions; this is the duty of the audit committee. The regulation also regulates the total number of followers of the audit committee. Namely, the minimum number of committee members is three people who come from public companies or external parties, independent commissioners. In addition, at least one audit committee member must have an educational background in accounting or finance (Hesniati 2021).

Companies with more audit members play a role in bringing experience, skills, and different views. This can be useful for carrying out adequate supervision of management, who handles company reports, which can help companies to maximize the company's annual reports that report intellectual capital disclosures (Ahmed Haji 2015; Taliyang and Jusop 2011). With a large number of committee members, this can minimize the company's agency costs because the audit committee's role in conducting supervision is more effective. Therefore, the larger the audit committee in a company. The greater the possibility of information submitted by the company in the annual report. With a large size, the power of the audit committee is also more significant. Hariyanto (2022), in their research revealed that the audit committee affected the disclosure of intellectual capital. The results of research by (Appuhami and Bhuyan 2015; Hesniati 2021; Taliyang and Jusop 2011) support the results of research previously conducted by (Baldini and Liberatore 2016).

**Hypothesis 1 (H1).** *There is an influence between the size of the audit committee and the disclosure of intellectual capital.*

### 2.4. The Effect of the Number of Audit Committee Meetings with Intellectual Capital Disclosure

Regarding the effect of the Number of audit committee meetings on the disclosure of intellectual capital, POJK Regulation No. 55/POJK.04/2015 states that meetings can be held periodically with a quarterly minimum, i.e., four times within a year. For example, if half of the members are present at the meeting, a meeting between members of the audit committee will be held. On the other hand, Li et al. (2008) recommend that committee meetings be held at least three to four times a year. Meetings are held so that tasks can be

carried out effectively in supervising two reports, namely financial and annual, as well as internal control and corporate governance (Baldini and Liberatore 2016).

Committee member meetings with a reasonably frequent intensity aim to monitor company reports to be more effective and be able to evaluate notifications that should be sent to people who use reports, namely by disclosing intellectual capital (Gamayuni 2015). The more often members of an audit committee hold meetings, the greater the amount of information that can be evaluated by the audit committee related to aspects that can affect the supervision of the process of making company reports that are effective and efficient. This is because much can be discussed at each meeting. Therefore, a high total number of audit committee meetings can increase the supervision of the audit committee; the more frequent the supervision by an audit committee, the less information gap there is between agents and principals. This is in line with the elaboration of agency theory. Li et al. (2008), in their research, reveal an influence between the financial expertise of the audit committee and the disclosure of intellectual capital. Indarti et al. (2021), who conducted almost the same research as Ahmed Ahmed Haji (2015), also support this research. The results of this study are identical to previous studies.

**Hypothesis 2 (H2).** *The Number of audit committee meetings with intellectual capital disclosure.*

*2.5. The Effect of the Audit Committee's Financial Proficiency and Intellectual Capital Disclosure*

Regarding the effect of financial expertise on intellectual capital disclosure. The benefits obtained from the financial expertise that members of the audit committee are in the ability to convey the information needed by stakeholders and identify problems in financial reporting. This information provides impetus to the company in presenting high-quality disclosure of intellectual capital. Audit members with an accounting education background better recognize the form of financial statements by existing criteria. Having members who understand these financial statements can help other members minimize information asymmetry between agents and principals. What is stated in agency theory helps reduce agency costs.

Audit committees whose members are accounting and finance education graduates usually know the implications of the capital market when preparing quality intellectual capital disclosures (Ahmed Haji 2015). In communicating information related to creating company value, the audit committee must have an understanding that leads to increasing disclosure of intellectual capital. The Number of audit members who are graduates of financial or accounting education can increase the disclosure of intellectual capital because, with someone who is an expert in the field, it will be easier to understand the market needs for information regarding intellectual capital.

**Hypothesis 3 (H3).** *There is an influence between the audit committee's financial proficiency and intellectual capital disclosure.*

*2.6. The Effect of Intellectual Capital Performance on Intellectual Capital Disclosure*

In stakeholder theory, it is stated that the excellent performance of the company's intellectual capital is an excellent disclosure of intellectual capital (Sudibyo and Basuki 2017). Therefore, the better the company that has intellectual capital performance, the higher the level of disclosure (sound). This disclosure can increase the confidence of stakeholders and the company's good name in the eyes of the public. Idie Widigdo (2013), in their research, revealed that there was a significant influence between intellectual capital performance and intellectual capital disclosure (ICD).

**Hypothesis 4 (H4).** *There is an influence between intellectual capital performance and intellectual capital disclosure.*

The following framework can describe the relationship between variables in Figure 1.

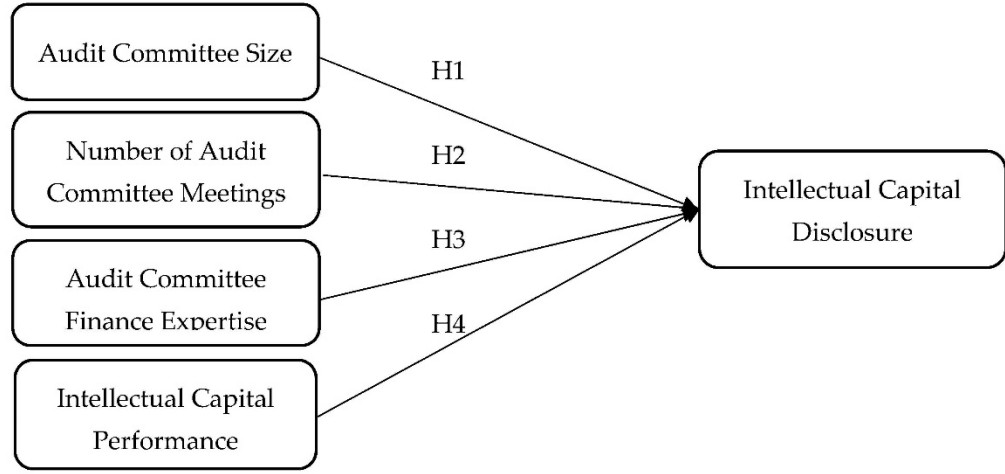

**Figure 1.** Research Framework.

### 3. Methodology

The researcher used a sample of financial service companies included in the IDX section in 2019–2021 with specific standards. Purposive sampling in this research was used as a way to obtain samples. Purposive sampling is a way to determine samples that meet specific standards and are considered representative. This method has a goal, namely, to obtain a representative sample that matches the researcher's standards. The following is the standard for determining the sample used in this study:

1.  Financial services companies in the IDX section periodically issue annual and financial reports for three consecutive years on the IDX from 2017 to 2019.
2.  The completeness and clarity of the data owned by a company are related to the research that the researcher did before.
3.  In the 2019–2021 period, the company posted no loss.
4.  The researcher used the annual report of service companies that are part of the IDX in the 2019–2021 period, which comprehensively includes the information needed as a secondary data source. In addition, researchers took data from the IDX website www.idx.co.id (accessed on 21 June 2022).
5.  The following are the results of the determination of the sample by the standard of determining the sample used in this study.

Based on Table 1, 91 companies listed on the Indonesia Stock Exchange from 2019–2021 are financial services companies. After taking samples that met the criteria determined by the researchers, the samples studied were 38 companies, so a total sample of 38 companies for three years obtained 114 observations and data that matched the criteria that could be processed in the 2019–2021 observation period were 114 observations.

**Table 1.** Sampling results.

| No | Information | Total |
|---|---|---|
| 1. | Total financial services companies from 2019–2021 | 91 |
| 2. | Financial services companies that do not issue annual reports and financial reports for three consecutive years (2019–2021) | (15) |
| 3. | Companies that do not meet the criteria and do not have the data used in the study | (16) |
| 4. | Companies that have suffered losses for 3 consecutive years (2019–2021) | (22) |
| 5. | Total research sample | 38 |
| 6. | Total observations (38 × 3 years) | 114 |

Source: data processed, 2022.

Components of intellectual capital disclosure; 22 components in Table 2. The independent variable in this study is intellectual capital. The measurement of intellectual capital itself uses three proxies: Value added capital coefficient (VACA)—VACA is the ratio between value added (V.A.) and working physical capital (C.A.). This ratio is an indicator of V.A. created by one unit of physical capital, human capital efficiency (VAHU). VAHU is how much is spent by workers from V.A. The relationship between V.A. and human capital (H.C.) indicates the ability of H.C. to create value in a company. Thus, the relationship between V.A. and H.C. shows the ability of H.C. to create value in the company; the structural capital coefficient (STVA) shows the contribution of structural capital (S.C.) in creating value. Variable operational definition in Table 3:

**Table 2.** Components of intellectual capital disclosure; 22 components.

| Category | Disclosure Items |
|---|---|
| Human Capital | 1. Number of employees<br>2. Education<br>3. Education and training<br>4. Employee competence<br>5. Employee turnover |
| Structural Capital | 6. Vision and mission<br>7. Code of conduct<br>8. Trademarks<br>9. Corporate governance<br>10. Organizational culture<br>11. Customer reporting system<br>12. Information system<br>13. Network system<br>14. Capital structure<br>15. Ability to pay the debt |
| Relational Capital | 16. Brand<br>17. Company name<br>18. Distribution network<br>19. Awards<br>20. Certification<br>21. Marketing strategy<br>22. Market Share |

Source: Ulum (2009); Masita and Muslih (2017).

**Table 3.** Variable operational definition.

| Variable | Definition | Indicator | Scale | Source |
|---|---|---|---|---|
| Audit committee size (SAC) (X1) | The SAC variable shows the Number of audit committee members in the company. | SAC = Number of audit committee members/3 | Ratio | (Das 2017) |
| Number of audit committee meetings (MAC) (X2) | The MAC variable shows the total meetings that audit committee members have held. | MAC = Number of audit committee Meetings/4 | Ratio | (Das 2017) |
| Audit committee finance expertise (FEXP_AC) (X3) | To find the value of the FEXP_AC variable, calculate the total number of committee members with the total Number of people who are experts in the field of finance. | Number of audit committee members with financial expertise/number of all members of the audit committee | Ratio | (Das 2017) |
| Intellectual capital performance (X4) | Intellectual capital performance shows the company's ability to manage and improve its I.C. This variable is measured using a measurement developed by Public, namely the value-added intellectual coefficient (VAICTM) method. | $VAIC_{TM} = VACA + VAHU + STVA$ | Ratio | (Das 2017) |

**Table 3.** *Cont.*

| Variable | Definition | Indicator | Scale | Source |
|---|---|---|---|---|
| Intellectual capital disclosure (Y) | Disclosure of intellectual capital concepts can provide resources in the form of new knowledge or knowledge that contains information about the company's intangible value and describes intangible assets that can be used to increase company value. | Intellectual capital disclosure = Number of items disclosed/total disclosure of intellectual capital. | | (Hesniati 2021) |

Source: data processed (2022).

*Dependent Variable*

Disclosure of intellectual capital is the dependent variable used by researchers in this study. Disclosure of intellectual capital is a design that can convey resources in the form of new knowledge that contains data on the company's intangible value and defines intangible assets that can be used to increase the company's value (Masita and Muslih 2017). In this study, disclosure of intellectual capital uses the ICD framework developed by several previous studies, including research by (Ferreira et al. 2012). In this study, researchers took 22 items used as benchmarks. In their research, (Tran et al. 2020) modified the ICD framework by using the scheme to increase intellectual capital disclosure (Sveiby 1997). First, the researcher used a numerical code to identify each component's disclosure of intellectual capital. The numbers used for the code were zero (0) and one (1). 0 was used for components that the company did not disclose, and 1 was used for components that were disclosed. Furthermore, after the Number of components of intellectual capital disclosure disclosed by the company was known, the amount owned by the component was divided by the number of components of intellectual capital, which consisted of 22 criteria components in the study.

## 4. Results and Discussion

Table 4 shows that the audit committee's (SAC)'s variable size has a minimum value of 1.00, which explains that the total number of audit committee members is at least three. 55/POJK.04/2015 states that the minimum Number of members of the audit committee in a company is three people. The maximum size of the audit committee has a value of 2.33 or a maximum of seven people, which was found at P.T. Bank Rakyat Indonesia (Persero) Tbk. 2019. The average value of the audit committee size from 114 observation samples was 1.14809, with a standard deviation of 0.274667.

**Table 4.** Descriptive statistics test results.

| | N | Minimum | Maximum | Mean | Std. Deviation |
|---|---|---|---|---|---|
| SAC | 114 | 1.000 | 2.333 | 1.14809 | 0.274667 |
| MAC | 114 | 1.000 | 5.500 | 2.12269 | 1.090258 |
| FEXP_AC | 114 | 0.250 | 1.000 | 0.62848 | 0.233525 |
| VAIC | 114 | 0.929 | 6.239 | 2.99112 | 1.131761 |
| ICD | 114 | 0.682 | 1.000 | 0.84444 | 0.081415 |
| Valid N (listwise) | 114 | | | | |

Source: data processed by SPSS 26 (IBM, Chicago, US), 2022.

*4.1. Normality Test*

This study uses the Kolmogorov–Smirnov test as the normality test, which is used to increase the value of the data normality test results. The condition for the data to be generally distributed in the Kolmogorov–Smirnov test is that the *p*-value must exceed 0.05. If the p-value is below 0.05, the data used is not normally distributed. The following is a table of normality test results in Table 5.

**Table 5.** Normality test results of one-sample Kolmogorov–Smirnov test.

|  |  | Unstandardized Residual |
| --- | --- | --- |
| N |  | 114 |
| Normal Parameters | Mean | 0.0000000 |
|  | Std. deviation | 0.05057782 |
| Most extreme differences | Absolute | 0.096 |
|  | Positive | 0.096 |
|  | Negative | −0.041 |
| Kolmogorov–Smirnov Z |  | 0.995 |
| Asymp. Sig. (2-tailed) |  | 0.275 |

Source: data processed using SPSS 26 (IBM, Chicago, US), 2022.

Based on Table 5, the residual regression equation test results show a value of 0.275 for their significance probability, which succeeded in exceeding the value of 0.05. Thus, the data used in this study are normally distributed.

*4.2. Multicollinearity Test*

A multicollinearity test is performed to clarify whether there is a correlation between variables in the regression model. Seeing the value of VIF in the regression model helps determine whether there is a correlation in this study. The occurrence of multicollinearity can be observed through the tolerance value; if the tolerance value is lower, the VIF value will be high (VIF = 1/tolerance). The cut-off value used aims to describe the existence of multicollinearity, namely VIF 10. A good regression reference is a regression model that does not have multicollinearity problems with its independent variables. Table 6 shows the results of the multicollinearity test. The result of the calculation of the tolerance value shows that there is no independent variable that has a value below 0.10. For example, the audit committee size variable of 0.975, the number of audit committee meetings at 0.925, the audit committee's financial expertise variable of 0.938, and the intellectual capital performance variable of 0.982. Based on the results of this tolerance, the research data do not occur as multicollinearity between the independent variables.

Then, from the results of the VIF calculation, it was also found that no independent variable had a VIF value of more than 10. This can be seen from the test results in the coefficients table, namely for the audit committee size variable of 1.026; the variable Number of audit committee meetings at 1.081; the financial expertise variable of 1.069, and the intellectual capital variable of 1.018. With the results of this calculation, in these research data, there is no multicollinearity between independent variables in the regression model.

**Table 6.** Multicollinearity rest results.

| Model | Unstandardized Coefficients | Standardized Coefficients | T | Sig. | Collinearity Statistics | |
|---|---|---|---|---|---|---|
| | B | Std. Error | Beta | | | Tolerance | VIF |
| (Constant) | 0.282 | 0.015 | | 18.822 | 0.000 | | |
| SAC | 0.057 | 0.024 | 0.219 | 2.424 | 0.017 | 0.975 | 1.026 |
| MAC | 0.022 | 0.006 | 0.328 | 3.549 | 0.001 | 0.925 | 1.081 |
| FEXP_AC | 0.019 | 0.028 | 0.062 | 0.675 | 0.501 | 0.938 | 1.069 |
| VAIC | 0.002 | 0.005 | 0.032 | 0.355 | 0.723 | 0.982 | 1.018 |

Source: data processed using SPSS 26 (IBM, Chicago, US), 2022.

*4.3. Autocorrelation Test*

The autocorrelation test's function is to determine whether there is a correlation between the t-period confounding error and the previous period (t − 1). Autocorrelation problems occur if there is a correlation between them. In this study, the Durbin–Watson test was used to test the presence or absence of autocorrelation. The criteria for deciding that there is no autocorrelation are du < d < 4 − du. The following table shows the results of the autocorrelation test.

Based on Table 7, the value of Durbin–Watson (D.W.) in this study is 1.776, compared with the value in the table that uses a significance of 0.05. According to the table with the Number of samples (n) 114 and the Number of independent variables (k) four, the D.W. value is du = 1.7637. Thus, the value of D.W. in the study is more significant than du, namely 1.7637 and smaller than 4 − du, namely 4 − 1.7637 = 2.2383, or 1.7637 < 1.776 < 2.2383. Based on these results, the regression model in this study does not experience autocorrelation between the independent variables.

**Table 7.** Autocorrelation test results.

| Model | R | R Square | Adjusted R Square | Std. the Error in the Estimate | Durbin–Watson |
|---|---|---|---|---|---|
| 1 | 0.438 | 0.192 | 0.160 | 0.051560 | 1.776 |

Source: data processed using (IBM, Chicago, US), 2022.

*4.4. Heteroscedasticity Test*

This test aims to prove the occurrence of residuals between observations in the model, and the occurrence of variance mismatches is the purpose of this test. Homoscedasticity occurs because the variation between the residuals of one observation and another does not change. On the contrary, if it changes, it is heteroscedastic. The heteroscedasticity test that the researcher used is the geyser test. The way to test this is by sorting the absolute value of the unstandardized residual, such as the dependent variable, backward (regression). For example, if all the independent variables are > 0.05, heteroscedasticity does not occur in the regression model. The results of the heteroscedasticity test can be seen in Table 8.

**Table 8.** Heteroscedasticity test results.

| Model | Unstandardized Coefficients | | Standardized Coefficients | T | Sig. |
|---|---|---|---|---|---|
| | B | Std. Error | Beta | | |
| (Constant) | 0.043 | 0.010 | | 4.457 | 0.000 |
| SAC | −0.018 | 0.015 | −0.115 | −1.162 | 0.248 |
| MAC | 0.001 | 0.004 | 0.027 | 0.268 | 0.789 |
| FEXP_AC | 0.017 | 0.018 | 0.097 | 0.964 | 0.337 |
| VAIC | −0.002 | 0.003 | −0.051 | −0.513 | 0.609 |

Source: Data processed using SPSS (IBM, Chicago, US), 2022.

Based on Table 8, no independent variable has a significance value below the alpha value of 0.05. The results of the heteroscedasticity test can be seen in table Sig. Namely, the variable size of the audit committee is 0.248, the variable Number of audit committee meetings is 0.789, and the financial expertise variable is 0.337. Furthermore, the intellectual capital performance variable is 0.609. With the test results above, the data in this study did not experience heteroscedasticity between independent variables in the regression model.

The results of the coefficient of determination can be seen from the model summary table in the adjusted R square section. For example, in Table 9, the value of the adjusted R square states that the coefficient of determination is 0.160 or 16%. Based on this value, it can be explained that the variable size of the audit committee, the Number of audit committee meetings, the financial expertise of the audit committee, and the performance of intellectual capital have a simultaneous influence on the dependent variable of intellectual capital disclosure by 16%. The other 84% are explained by independent variables not included in this research.

**Table 9.** Coefficient of determination test results.

| Model | R | R Square | Adjusted R Square | Std. the Error in the Estimate |
|---|---|---|---|---|
| 1 | 0.438 | 0.192 | 0.160 | 0.051560 |

Source: Data processed by SPSS 26 (IBM, Chicago, US), 2022.

*4.5. Significance Test (F Test)*

This test aims to determine whether all the independent variables used in the regression model have a combined effect on the dependent variable. The results of the F test can be seen in the table below (Table 10).

**Table 10.** F-test results.

| Model | Sum of Squares | Df | Mean Square | F | Sig. |
|---|---|---|---|---|---|
| Regression | 0.064 | 4 | 0.016 | 6.046 | 0.000 |
| Residual | 0.271 | 102 | 0.003 | | |
| Total | 0.335 | 106 | | | |

Source: data processed using SPSS 26, 2022.

Table 10 explains that the significant value or *p*-value is 0.000. This result shows that the *p*-value is not greater than the alpha value (0.005). Thus, the model used is correct.

*4.6. Multiple Linear Regression Test*

In this study, the analysis used is a multiple linear regression analysis model. The usefulness of the analysis is in testing two or more independent variables on the dependent variable. The dependent variable that the researcher use is the disclosure of intellectual

capital. The independent variable in this study has two characteristics: the audit committee and intellectual capital performance. In addition, the size of the audit committee, the financial expertise of the audit committee, and the total number of audit committee meetings are the characteristics of the committee used in this study. The following are the results of the multiple regression test, which are shown in the table below.

Based on Table 11, the relationship between the characteristics of the audit committee and the performance of intellectual capital with intellectual capital disclosure can be made a regression equation:

$$Y = 0 + 1X1 + 2X2 + 3X3 + 4X4 + e$$

$$ICD = 0 + 1 \text{ Committee size} + 2\text{Number of meetings} + 3\text{Financial expertise} + 4\text{IC performance} + e$$

$$ICD = 0.282 + 0.057 \text{ Committee size} + 0.022 \text{ Number of meetings} + 0.019 \text{ Financial expertise} + 0.002 \text{ IC performance} + 0.15$$

Discussion of Data Analysis Results.

**Table 11.** Regression test results.

| Model | Unstandardized Coefficients | | Standardized Coefficients | T | Sig. |
|---|---|---|---|---|---|
| | B | Std. Error | Beta | | |
| (Constant) | 0.282 | 0.015 | | 18.822 | 0.000 |
| Audit committee size | 0.057 | 0.024 | 0.219 | 2.424 | 0.017 |
| Number of audit committee meetings | 0.022 | 0.006 | 0.328 | 3.549 | 0.001 |
| Audit committee finance expertise | 0.019 | 0.028 | 0.062 | 0.675 | 0.501 |
| Intellectual capital performance | 0.002 | 0.005 | 0.032 | 0.355 | 0.723 |

Source: data processed using SPSS 26, 2022.

### 4.7. Audit Committee Size on Intellectual Capital Disclosure

Based Table 12, the influence of the audit committee on intellectual capital disclosure is the first hypothesis in this study. The results of this study are in line with the results of research by (Ahmed Haji 2015; Hariyanto 2022; Hesniati 2021; Indarti et al. 2021). According to (Taliyang and Jusop 2011), the size of the audit committee influences the disclosure of intellectual capital. This is also supported by the audit committee size on intellectual capital disclosure (Li et al. 2008), which explains that the more members on the audit committee there are, the more quality control supervision over the accounting and financial processes of the company will increase. This occurs because of the diversity of skills, views, and experiences of each member in ensuring adequate supervision of company reports. This further increases the disclosure of intellectual capital that the company presents in the annual report.

**Table 12.** Hypothesis testing results.

| No | Hypothesis | Results | Conclusion |
|---|---|---|---|
| H1 | Audit committee size → Intellectual capital disclosure | t = 2.424, Sig. 0.017 < 0.05 | **Accepted** |
| H2 | The number of audit committee meetings → Intellectual capital disclosure | t = 3.549, Sig. 0.001 < 0.05 | **Accepted** |
| H3 | Audit committee finance expertise → Intellectual capital disclosure | t = 0.675, Sig. 0.501 > 0.05 | **Rejected** |
| H4 | Intellectual capital performance → Intellectual capital disclosure | t = 0.355, Sig. 0.723 > 0.05 | **Rejected** |

Data source: data processed, 2022.

Adequate supervision of company reports carried out by audit committee members is in line with the theory of stewardship, where management is afforded the trust to work well for the benefit of the public and stakeholders (Buallay and Al-Ajmi 2019). Audit committee members help management to monitor and provide the best results for the company. With more and more audit committee members working effectively, the intellectual capital owned by the company runs effectively. This encourages management to disclose more intellectual capital owned by the company.

### 4.8. Number of Audit Committee Meetings on Disclosure of Intellectual Capital

The second hypothesis in this study intends to determine the effect of the Number of audit committee meetings on the disclosure of intellectual capital. This study's results align with research conducted by (Hariyanto 2022), which states that the number of audit committee meetings influences intellectual capital disclosure. Frequent audit committee meetings can improve a company's reporting monitoring process to be more efficient and evaluate the information that needs to be submitted to report users, one element of which is information about the company's intellectual capital (Hesniati 2021). This is in line with agency theory. In addition, meetings that audit committee members hold often can reduce the information gap between agents and principals.

An audit committee with various kinds of experts who meet frequently discusses evaluation and implementation strategies, one of which is overseeing financial statements, corporate governance and internal control. The high frequency of audit committee meetings allows the company to increase the disclosure of intellectual capital owned by the company. Although this disclosure is made voluntarily, through regular meetings held by the audit committee, the audit committee can consider issues in business competition and the shift in business from labour-based to knowledge-based (Tjahjadi et al. 2019). Therefore, one of the main factors in increasing firm value is the disclosure of intellectual capital.

### 4.9. Audit Committee Financial Expertise on Intellectual Capital Disclosure

The influence of the audit committee's financial expertise on intellectual capital disclosure is the third hypothesis in this study. This study's results differ from the research conducted by (Ahmed Haji 2015), which states that financial expertise significantly influences intellectual capital disclosure. However, this study's results align with the research conducted Buallay and Al-Ajmi (2019) audit committee financial expertise on intellectual capital disclosure. Which explained that the financial expertise variable does not affect intellectual capital disclosure. Financial expertise is less relevant to disclosing intellectual capital but more relevant to discussing financial issues. This is because some elements of intellectual capital, such as corporate culture, company brands, and others, require special knowledge to understand them, not just financial expertise.

To increase the disclosure of intellectual capital, more than only the financial expertise of the audit committee is required to understand all elements of intellectual capital. Thus, the audit committee must have expert members with other skills. Such as management expertise, technology and information, and other expertise (Buallay and Al-Ajmi 2019). Thus, other experts are needed in the audit committee to increase the company's intellectual capital.

### 4.10. Intellectual Capital Performance on Intellectual Capital Disclosure

The last hypothesis in this study intends to determine the effect of intellectual capital performance on intellectual capital disclosure. This study's results differ from the results of research conducted by (Gamayuni 2015), which states that intellectual capital performance influences intellectual capital disclosure. The better the performance of intellectual capital owned by a company, the higher (good) the likelihood of the company disclosing intellectual capital.

The results of this study align with research conducted by (Widigdo 2013), which explains that there is no influence between the intellectual capital performance variable and

intellectual capital disclosure. This occurs because the disclosure of intellectual capital is voluntary. There are no detailed rules or regulations regarding the disclosure of intellectual capital in Indonesia, so companies do not try to increase the disclosure of intellectual capital owned by them in their annual reports. In this case, the stakeholder theory cannot be proven because management does not try to disclose intellectual capital, so the information obtained by stakeholders could be more optimal.

## 5. Conclusions

Based on this study's results and the description in Section 4, this study aimed to determine the effect of an audit committee's characteristics, the size of an audit committee, the Number of audit committee meetings, and the expertise of the audit committee on the audit committee and intellectual capital performance in terms of intellectual capital disclosure. Therefore, the conclusions that can be drawn from the results of multiple regression analysis are as follows:

The size of the committee can affect the disclosure of intellectual capital. This can occur because the more members of the audit committee there are, the more supervision is carried out on the company's accounting and financial processes. In addition, each member's enormous diversity of views, experiences, and skills can also increase the effectiveness of monitoring company reports, which helps optimize intellectual capital disclosures reported in the annual report.

The Number of audit committee meetings influences intellectual capital disclosure because the more meetings the audit committee holds, the more evaluations are discussed, which later need to be submitted to report users, one of which can be presented about the company's intellectual capital. On the other hand, the audit committee's financial expertise does not affect intellectual capital disclosure. This is because financial expertise is less relevant for disclosing intellectual capital but more relevant for discussing finance. This occurs because some elements of intellectual capital, such as corporate culture, information systems, trademarks, and others, require special skills and knowledge to be understood.

Intellectual capital performance does not affect intellectual capital disclosure. Intellectual capital performance is not influenced because there are no rules governing intellectual capital disclosure in Indonesia. This is a voluntary disclosure, so the companies' management departments need to optimize the disclosure of intellectual capital owned in their company reports.

Regarding this observation, several shortcomings can prevent the results of the observations from agreeing with the hypothesis made. Therefore, the limitations of this study can be reviewed for future research. The following are some of the limitations of this study: The research used only four variables, so the adjusted R square value is 16%, and other independent variables influence the remaining 84%. The content analysis method in this observation is also prone to the researcher's subjectivity as a scorer in intellectual capital disclosure.

Based on the limitations of the research above, the researchers suggest several suggestions for further research: Future researchers should be able to include other independent variables such as the independence of the audit committee, company size, quality of external auditors, and other variables that are thought to influence intellectual capital disclosure. In addition, it is recommended that for further research, researchers can be assisted by one other person to provide a score or value in the disclosure of intellectual capital to minimize subjectivity.

**Author Contributions:** Conceptualization, W.M.; methodology, H.M.; validation, M.H.; formal analysis, H.M.; investigation, H.M.; resources, M.H.; data curation, M.H.; writing-original draft preparation, M.H.; project administration, M.H.; funding acquisition, W.M.; writing-review & editing, W.M. All authors have read and agreed to the published version of the manuscript.

**Funding:** This research was funded by Research Funded Research of International Publications of High Reputation (RPIBT) Universitas Diponegoro, grant number No. SPK: 233-33/UN7.6.1/PP/2022

and the APC was funded by the Directorate of Research and Community Service, Ministry of Education, Culture, Research, and Technology, Indonesia.

**Data Availability Statement:** Not Applicable.

**Acknowledgments:** Research Funded By Research of International Publications of High Reputation (RRPIBT) Universitas Diponegoro No. SPK: 233-33/UN7.6.1/PP/2022.

**Conflicts of Interest:** The authors declare no conflict of interest.

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
