# Peer review of "The Role of Audit Committee Characteristics and I.C. Performance on I.C. Disclosure: Evidence from the Indonesian Banking Sector"

_economies, doi:10.3390/economies11010007_

Round 1

Reviewer 1 Report

·         Summary: the aim of the article is important for researchers in the field of intellectual capital since the disclosure of intangible assets has been interesting research for a long time. This study provides new evidence on this topic which is justified with a clear methodology and comparison with previous studies.

·         General concept comments

·         Article: the article is interesting and important to allow future research on the field. However, the English need to be improved so that some sentences can make sense and to improve the overall quality of the work presented.

·         Specific comments:

o   Introduction: it clearly presents the context for the study and its goal. However, some of the text namely the comparison between different studies should be integrated into the literature review section.

o   Literature review: why use the 3 dimensions instead of the 4 that are being used currently (which means including social capital). Need to explain the choice of the use of the IC dimensions. The model is separated into two different pages (number 4 and 5). Hypotheses are well justified.

o   Methods: which author was used to describe the importance of purposive sampling? In the literature review it mentions organisational capital and Table 2 uses Structural Capital – coherence on the use of concepts.

o   Conclusion: chapter 6 can be merged with chapter 5 as it follows the previous text. The conclusion is well done with limitations and future research suggestions included. The second paragraph would be better with bullet points for each idea.

o   References: duplication and errors  

§  Gamayuni, Rindu Rika. 2015b. “The Effect Of Intangible Asset Financial Performance And Financial Policies On The Firm Value.” International Journal of Scientific & Technology Research 4(1):202–12.

§  Hesniati, Hesniati. 2021a. “Effect of Corporate Governance on Intellectual Capital Disclosure.” International Journal of Economics, Busi-644 ness and Accounting Research (IJEBAR) 5(1):46–63. doi: 10.29040/ijebar.v5i1.1584. 645.

§  Li, Jing, Richard Pike, and Roszaini Haniffa. 2008a. “Intellectual Capital Disclosure and Corporate Governance Structure in UK Firms.” Accounting and Business Research 38(2):136–59. doi: 10.1080/00014788.2008.9663326.

§  Widiatmoko, J, M. G. K. Indarti, and I. D. Pamungkas. 2020. “Corporate Governance on Intellectual Capital Disclosure and Market Capitalization.” Cogent Business & Management 7(1):1750332. 687

Author Response

Thanks For Review and Information.

It is necessary to explain the choice of using IC dimensions. Here is our additional explanation: There is a need to go beyond IC reporting. Innovations in ICD, such as integrating reporting, disclosure in ecosystems, and stakeholder engagement, open up new possibilities for future research (Dumay, 2012). As does how to update and apply existing approaches to today's dynamic, knowledge-driven, intangible based organizations (Cuozzo et al., 2017), where comparability across companies, moving beyond a Euro-centric view of IC or helping investors find the suitable needles in the haystack of their information overload are critical. Tan, Plowman, and Hancock (2007) classify IC into three basic formats, namely: (1) human capital; (2) structural capital; and (3) customer capital. (Leliaert, Candies, and Tilmans 2003) developed the 4-Leaf model, which classifies IC into human, customer, structural capital and strategic alliance capital (Tan et al., 2007). Intellectual Capital referred to in this study is IC performance which is measured based on the value added created by physical capital (VACA), human capital (VAHU), and structural capital (STVA). The combination of the three added values is symbolized by the name VAIC™ which was developed by (Pulic, 1998).

Value added is the most objective indicator for assessing business success and shows the company's ability to create value (value creation). VA is calculated as the difference between output and input. Output (OUT) represents revenue and includes all products and services sold in the market, while input (IN) includes all expenses in obtaining revenue. The critical thing in this method is that labor expenses are not included in the IN. Due to its active role in the value creation process, intellectual potential (represented by labor expenses) is not counted as a cost and is not included in the IN component. Therefore, an essential aspect of Pulic's method is treating labor as a value-creating entity. VA is influenced by the efficiency of Human Capital (HC) and Structural Capital (SC). Another relationship of VA is capital employed (CE), which in this case is labeled VACA. VACA is an indicator for VA created by one unit of physical capital.

The independent variable in this study is intellectual capital. The measurement of intellectual capital itself uses three proxies: Value added capital coefficient (VACA). VACA is the ratio between value added (VA) and working physical capital (CA). This ratio is an indicator of VA created by one unit of physical capital, The Human Capital Efficient (VAHU) VAHU is how much rupiah workers spend from VA. The relationship between VA and Human Capital (HC) indicates the ability of HC to create value in a company. So the relationship between VA and HC shows the ability of HC to create value in the company, Structural Capital Coefficient (STVA) Shows the contribution of structural capital (SC) in creating value.

The models are separated into two different pages (numbers 4 and 5), we have edited and adjusted the model images. The hypothesis is well justified (we have adjusted and edited it).

In Conclusion: we have merged chapter 6 with chapter 5. The second paragraph would be better with bullet points for each idea we have revised. Thank you for the advice that has been given.

We have removed references: duplication and errors so there are no duplications and errors like:

  • Gamayuni, Rindu Rika. 2015b. “The Effect Of Intangible Asset Financial Performance And Financial Policies On The Firm Value.” International Journal of Scientific & Technology Research 4(1):202–12.
  • Hesniati, Hesniati. 2021a. “Effect of Corporate Governance on Intellectual Capital Disclosure.” International Journal of Economics, Busi-644 ness and Accounting Research (IJEBAR) 5(1):46–63. doi: 10.29040/ijebar.v5i1.1584. 645.
  • Li, Jing, Richard Pike, and Roszaini Haniffa. 2008a. “Intellectual Capital Disclosure and Corporate Governance Structure in UK Firms.” Accounting and Business Research 38(2):136–59. doi: 10.1080/00014788.2008.9663326.
  • Widiatmoko, J, M. G. K. Indarti, and I. D. Pamungkas. 2020. “Corporate Governance on Intellectual Capital Disclosure and Market Capitalization.” Cogent Business & Management 7(1):1750332. 687

Reviewer 2 Report

The paper is missing a lot of very important references for the field of IC disclosure. See for example:

Cuozzo, B., Dumay, J., Palmaccio, M., & Lombardi, R. (2017). Intellectual capital disclosure: a structured literature review. Journal of Intellectual Capital, 18(1), 9-28.

Dumay, J. (2012), “Grand theories as barriers to using IC concepts”, Journal of Intellectual Capital, Vol. 13 No. 1, pp. 4-15.

Moreover, while findings are well presented, a discussion section that ties together findings and literature review is missing.

Limitations and future perspectives of the study must be presented more accurately, as well as theoretical and practical implications.

Author Response

We have added the article and thank you for the suggestions given so that we can add support for this paper to be better, such as:

  1. Dumay, John C. (2012). "Grand Theories as Barriers to Using IC Concepts." Journal of Intellectual Capital.
  2. Cuozzo, Benedetta, John Dumay, Matteo Palmaccio, and Rosa Lombardi. 2017. “Intellectual Capital Disclosure: A Structured Literature Review.” Journal of Intellectual Capital 18(1):9–28.

Findings are well presented, a discussion section that ties together findings, we have joined together to make it even better. Thank you for the suggestions and reviews that have been given.
